# Effects of Visceralising Leishmania on the Spleen, Liver, and Bone Marrow: A Pathophysiological Perspective

**DOI:** 10.3390/microorganisms9040759

**Published:** 2021-04-05

**Authors:** Aikaterini Poulaki, Evangelia-Theophano Piperaki, Michael Voulgarelis

**Affiliations:** 1Department of Pathophysiology, School of Medicine, National and Kapodistrian University of Athens, 115 27 Athens, Greece; aikaterini.poulaki@gmail.com; 2Department of Microbiology, School of Medicine, National and Kapodistrian University of Athens, 115 27 Athens, Greece

**Keywords:** leishmaniasis, visceral leishmaniasis, immunobiology, liver, spleen, microenvironment, bone marrow dysplasia, pancytopenia

## Abstract

The leishmaniases constitute a group of parasitic diseases caused by species of the protozoan genus *Leishmania*. In humans it can present different clinical manifestations and are usually classified as cutaneous, mucocutaneous, and visceral (VL). Although the full range of parasite—host interactions remains unclear, recent advances are improving our comprehension of VL pathophysiology. In this review we explore the differences in VL immunobiology between the liver and the spleen, leading to contrasting infection outcomes in the two organs, specifically clearance of the parasite in the liver and failure of the spleen to contain the infection. Based on parasite biology and the mammalian immune response, we describe how hypoxia-inducible factor 1 (HIF1) and the PI3K/Akt pathway function as major determinants of the observed immune failure. We also summarize existing knowledge on pancytopenia in VL, as a direct effect of the parasite on bone marrow health and regenerative capacity. Finally, we speculate on the possible effect that manipulation by the parasite of the PI3K/Akt/HIF1 axis may have on the myelodysplastic (MDS) features observed in VL.

## 1. Introduction

The kinetoplastid *Leishmania* spp. is a unicellular parasitic microorganism that affects the mammalian reticuloendothelial system (RES) [1]. The genus encompasses more than 20 species causing three clinical syndromes, cutaneous (CL), mucocutaneous (MCL), and visceral (VL) leishmaniasis, which have several subdivisions each and exhibit overlap in clinical manifestations and causative species, particularly in immunocompromised hosts [2]. The complex interplay of parasitic and host factors and the resulting clinical variability render the understanding of pathophysiology and the treatment of leishmaniases most challenging. Approximately 700,000 of VL and 1,200,000 cases of CL are estimated to occur annually [3]. Of the infected patients, only a small fraction develop symptomatic disease [3].

The visceral form (VL) is a debilitating, chronic wasting syndrome caused by members of the *L. donovani* complex (*L. donovani* and *L. infantum*) in the Old World and *L. chagasi (syn. L. infantum)* in the New World. Nearly all patients suffering from symptomatic disease succumb to VL-related complications within two years if left untreated [1,3,4], while the disease is universally fatal in immunocompromised hosts. The asymptomatic to symptomatic ratio of infected individuals varies greatly among endemic countries (from 1:2.4 in Sudan to 50:1 in Spain) [5]. This wide variability most likely reflects the population’s health status with socioeconomic factors and acquired immunodeficiencies, mostly HIV infection, decreasing the ratio. An inherent variability of the immune response in different populations also exists [5]. The therapeutic options are limited to a few relatively toxic classical drugs, with a profound paucity of newer agents. A cure is far from guaranteed since *Leishmania* eradication requires an effective immune response, so the parasite persists in the immunocompromised patients [2].

In subjects with asymptomatic VL and a Th1-polarized T helper immune response that primes macrophages toward a leishmanicidal M1 phenotype, it efficiently clears the parasitic load [6,7]. Self-healing is associated with the regulated production of proinflammatory cytokines, mostly INF-γ, IL-12 and ΤΝF-α [8], highlighting the pivotal role of cell-mediated response in effective immunity [9,10]. Despite continuing research, the immunobiology of symptomatic disease is still poorly understood. While all *Leishmania* species are capable of immunologic manipulation of their host, only visceralising ones possess the ability to survive and disperse inside the mammalian host, infecting its internal organs [11,12]. These species reach the spleen and initiate an immunologic cascade of overwhelming dendritic cell activation, loss of stromal cells, complete splenic disorganization, and priming of macrophages to a disease-amplifying phenotype [6,8,13,14,15,16]. In contrast, the liver is more efficient in clearing the intruders [6]. Over 70% of the parasitic load is cleared from the hepatic parenchyma by granuloma formation [6,8]. The biology of this marked distinction between the two organs has not been clearly explained.

The bone marrow (BM) is another organ seriously affected by parasitic infestation [17], with pancytopenia completing the VL defining triptych of splenomegaly and hypergammaglobulinemia [1]. Although reduction of platelets (PLTs) and red blood cells (RBCs) is attributed to peripheral destruction and increased pooling, an in-depth investigation of the pathophysiologic processes underlying pancytopenia has only recently begun [18,19,20]. The natural course of VL is characterized by delayed correction of cytopenias and BM phenotypic changes, suggesting that *Leishmania* directly affects the BM in a manner resembling acquired myelodysplastic syndromes (MDS) (Figure 1) [12,21,22,23].

In this review, we outline the immunobiology of splenic failure and hepatic success in dealing with the parasite, and we attempt to elucidate the effects of visceralising *Leishmania* on the hematopoietic system based on the existing literature. The potential pathophysiological mechanisms involved in the VL-associated pancytopenia are also discussed [21]. It should be borne in mind that almost all existing literature derives from non-human mammal models, mostly murine [8,24]. Regarding the mechanisms discussed below, we attempt to clearly specify whether the data derives from an animal, a human, or an in vitro model, but in any case, extrapolation to the human host should be attempted with caution.

## 2. Immunobiology of Hepatic versus Splenic Leishmanial Responses

### 2.1. Overview of the Leishmanicidal Response in the Liver

During this initial phase, which corresponds to the second–third week post-infection (p.i.) in VL murine models, most parasites are cleared from the liver parenchyma by the hepatic immunologic response [6,16,25]. In other mammalian models, namely dogs and macaques, the presence of infected reticuloendothelial liver cells (Kupffer cells) can be observed as early as 2 days p.i. and can indefinitely expand to months or years [26]. Briefly, with the help of both CD4+ Th1 and CD8+ cytotoxic cell responses, Kupffer cells (KCs) that are either infected or gather toward the core of the granuloma from adjacent sinusoids transform to the activated M1 phenotype and kill internalized parasites mainly through nitric oxide synthase (iNOS/NOS2) [8]. KCs are at the core of the granuloma and form the meshwork in which its maturation occurs. This reaction that, in most immunocompetent VL in vivo models suffices to confer parasitic clearance and a degree of tissue-specific memory, is a granulomatous response within the hepatic microenvironment, orchestrated by the KC [26,27]. A subset of T cells that plays a vital role in the effectiveness of the newly formed granuloma is the invariant natural killer T cells (iNKTs), which can secrete both INF-γ (immunostimulatory and modulatory) and IL-4 (immunosuppressive cytokines), depending on their cytokine input [28,29]. Physiologically, these cells react to glycolipid antigens presented through the CD1d surface receptor [30]. Although it has been experimentally shown in animals that iNKTs facilitate the effective liver granulomatous response, some studies suggest the opposite [28,31]. This discordance could be explained by the observation that excessive stimulation of iNKTs leads to anergy [28,32]. Therefore, the variable response of iNKTs, which are essential for the initiation of the granulomatous response, depending on the cytokine mixture they are exposed to in their microenvironment, may partially account for the differential immune response observed in different organs of the host.

Following KC and iNKT activation, the secretome of the developing granuloma attracts CD4+ and CD8+ T cells that facilitate further granuloma maturation [26]. Furthermore, parasites that are released from dying KC are taken up by liver dendritic cells (DCs). Upon contacting and internalizing the parasite, DCs maturate, reduce their phagocytic activity, and acquire a professional antigen-presenting function, characterized by increased MHC II expression and IL-12/ tumor necrosis factor-alpha (TNF-α) secretion [7,8,33].

Both T cell subtypes are indispensable for effective immune response and parasite clearance. Under the influence of this cytokine mixture, bystander and attracted CD4+ T helper cells polarize toward the Th1 phenotype and begin secreting [8]. Of the cytokines produced by CD4+ Th cells, ΙΝF-γ activates the effector M1 macrophage phenotype that produces large amounts of TNF-α in a self-stimulatory loop, that in this microenvironment drives a granulomatous immune response [8,34]. Hepatic stellate cells regulate the immune response through anti-inflammatory cytokines, mostly IL-10 and transforming growth factor-beta (TGF-β), which are produced either in direct response to infection or to increased TNF-α in the surrounding microenvironment [8,33,35] (Figure 1).

Several non-human mammalian models have proved the granuloma is so effective in eradicating the parasites that there is a resolution of infection upon adequate granuloma organization. This could also indicate that splenic dissemination in the presence of an effective liver response occurs during the initial parasitemia, namely on the second day p.i. in mice [25,26,29]. Epithelioid granulomas in various stages of maturation have been found in human liver and spleen samples from hosts with asymptomatic infections [36]. On the contrary, at almost 24 weeks p.i. absence and/or poor organization of liver granulomas translates to increased parasite burden and poor prognosis for the host [26]. An impressive distinction in the hepatic vs. splenic response is almost uniformly observed in murine models that constitute the bulk of experiments performed [26]. While the liver forms granulomas that effectively eradicate the parasite from as early as the fourth day p.i., the spleen is almost always overwhelmed by the infection [26].

### 2.2. Overview of the Ineffective Splenic Response

The importance of granuloma formation, a response that stands at the intersection of innate and adaptive immunity, is evident in humans with active VL. Splenomegaly, lymphadenopathy, and varying degrees of hepatomegaly are the hallmarks of visceralisation in leishmaniasis [13,16] and confirm the failure of the host’s granulomatous responses to control the parasitic load and thus prevent the establishment of the infection. The stark contrast between the hepatic and the splenic response can be explained to some extent by the fact that activation of macrophages, the host cells of *Leishmania*, takes place within a different microenvironment [37]. Histologically, splenic failure progresses in four sequential stages, which overlap significantly [16]. Since these models have been established using a mixture of human, canine, and rodent splenic tissues, any extrapolation to humans should be attempted with care.

The spleen can rapidly produce nonspecific, polyreactive antibodies, which efficiently contain infectious agents until the antigen-specific immune reaction occurs in the germinal center (GC). This essential survival trait, which is afforded by short-lived plasma cells residing in its marginal zone (MZ), is hijacked by visceralising *Leishmania* spp., leading to complete splenic anatomical and functional disorganization [8,38]. Initially, there is an expansion of the white pulp, in which there are increased numbers of activated macrophages surrounding hyperplastic lymphoid follicles with increased marginal zones (MZ) [15,16]. Upon initial contact with the parasite, marginal zone macrophages (MZM) present parasitic promastigote antigens [8,33]. This would normally lead to TNF-α production from macrophages and an effective killing response [7,8]. However, contrary to what occurs with other microorganisms, internalization of visceralising *Leishmania* through TLR-mediated PAMP (pathogen associated molecular pattern) recognition fails to elicit the appropriate TNF-α production in macrophages [8,39,40,41], possibly due to an effect *Leishmania* promastigotes have on the hypoxia-inducible factor 1 (HIF1) regulatory system (further discussed later) [42]. Research has shown that upon stabilization of HIF1 in naive macrophages, M2 regulatory polarization occurs [43,44]. The role of HIF1 in VL is summarized in Figure 2 and also further discussed in the BM section.

As expected, marginal zone B cells begin producing large amounts of nonspecific, polyreactive IgM antibodies (Ab) [8,14]. Opsonization of the parasite with these Ab facilitates its phagocytosis by splenic DCs, the resident professional antigen-presenting cells (APCs) [45], but due to poor Ab specificity, DCs fail to mature and migrate to regional lymph nodes and/or initiate the GC reaction [33,46]. At the same time, both splenic DCs that have phagocytosed opsonized parasites and MZ B cells secrete IL-10, a cytokine that directs monocyte maturation toward a regulatory M2 phenotype, further dampening macrophage response [8,47].

Splenic DCs are significantly more sensitive to antigenic stimulation than liver DCs, and after maturation, they produce increased amounts of proinflammatory cytokines, mainly TNF-α and B cell-activating factor (BAFF) [8,48].

The increased proinflammatory pressure attracts bone marrow-derived immature myeloid cells that home in on the splenic microenvironment and, as splenic myeloid cells, begin secreting large amounts of TNF-α as well [41,49,50]. Increased TNF-α leads to loss of MZ stromal cells and reduces their expression of key homing receptors, CCL19/CCL21, which are required for the migration of both T cells and DCs from the MZ toward the PeriArteriolar Lymphoid Tissue (PALS), a process that is crucial for their effective differentiation [8,51]. The concomitant hypersecretion of IL-10 by both DCs and M2 macrophages further downregulates the CCL19 ligand and CCR7 expression on the DCs as a reflex compensatory reaction to TNF-α [8,47].

The inflammatory state that is elicited, either by *Leishmania* or by other pathogens, drives the release of immature myeloid lineage from the BM [49,52]. These cells, as we explain later, are conditioned by the BM microenvironment and/or by their infection with Leishmania to become myeloid-derived suppressor cells (MDSC) in the spleen [52,53]. They induce a strong anti-inflammatory response, with increased amounts of IL-10 and TGF-β, further dampening an effective immune response and perpetuating the infection [7,49,52,53,54,55]. DCs residing in the disorganized, inflammatory, and markedly hypoxic splenic environment show increased HIF1 stabilization, which hampers their ability to produce the proinflammatory cytokine IL-12 and diverts them further toward IL-10 secretion [56,57,58,59].

Without the stromal cell meshwork to support MZ and PALS integrity, further destabilization of the lymphoid follicle occurs [16,51]. At this point, after the fourth week of infection, the spleen presents a deregulated architecture, with dilated erythroid channels, absence of definitive red-white pulp borders, and scattered small, ill-defined follicles with diminished MZ [14,16]. The remaining DCs produce BAFF, leading to increased homing and survival of mature IgG-producing long-lived plasma cells. These are drawn to the spleen from the periphery and BM and secret large amounts of Ab, causing the polyclonal hypergammaglobulinemia observed in VL [8,14,16,46]. Their relocation and expansion further damage splenic architecture, transforming it into a plasma cell survival niche [16]. The whole splenic immune response is summarized in Figure 1.

The disrupted microenvironment also has a pronounced effect on macrophage polarization [15]. Cytokine profiling of splenic macrophages reveals increased production of both pro- and anti-inflammatory cytokines. Notably, the increase in INF-γ observed both in the spleen and the liver suggests that increased INF-γ levels are not sufficient to prime an M1 effector phenotype, thus underlining the dominant role the microenvironment plays on cellular conditioning [53]. In chronic VL, stimulation of splenic macrophages with IFN-γ induces IL-10 production through STAT-3 activation, thus maintaining the disease-promoting M2 phenotype [53]. STAT-3 activation could additionally occur through the PI3K-Akt-mTOR pathway, which is also active in chronic VL. It has been shown that IL-10 can be induced through the PI3K/Akt pathway in infected myeloid cells [38,60,61,62]. Furthermore, as shown by Beattie L et al., the transcriptional profile of infected KCs differs significantly from that of uninfected KCs, which can effectively kill the parasite. An important pathway identified as being manipulated by the parasite was the retinoid X receptor alpha (RXRa) network, whose functions include regulation of lipid metabolism [63]. This network is known to indirectly activate the PI3K/Akt pathway through lipid modulation [64]. Thus, activation of the PI3K-Akt pathway appears to have multiple triggers and shows potential as a therapeutic target [65].

To conclude, regarding the immunopathology of VL, it should be borne in mind that much remains unknown, especially in human subjects. Furthermore, reaction to the parasite varies between individuals due to the inherent variability of immune responses. More than 90% of infected subjects clear the parasite in a subclinical fashion through granuloma formation, maturation, and resolution in both the liver and the spleen [26,41]. Although several cytokines can induce or inhibit this effective response, many of which are mentioned here, the etiology of this heterogeneity remains elusive [66,67].

## 3. Pathophysiology of Pancytopenia in VL

Peripheral blood cytopenia with variable involvement of the three hematopoietic lineages, the megakaryocytic, erythroid, and to a much lesser extent, the myeloid, is a hallmark of chronic inflammation [68]. VL, a parasitic infection of both the BM and the spleen, is commonly associated with pancytopenia [19] of unclear etiology [17,18,20].

### 3.1. Pathophysiology of Thrombocytopenia

Increased peripheral consumption with an inadequate BM response circumscribes the pathophysiology of VL-associated thrombocytopenia [18]. Several studies have failed to establish statistically significant differences in thrombopoietin (TPO) levels between kala-azar patients with thrombocytopenia and controls, indicating a lack of stimulation of the BM to produce platelets [69,70]. Histologically, defective TPO production is inferred by the relative cytoplasmic immaturity of differentiating megakaryocytes [69]. Coating of platelets with polyreactive antibodies that are produced by polyclonal B cell expansion and blood pooling in the expanded spleen, where they are phagocytosed by macrophages and dendritic cells, account for decreased peripheral PLT survival [18].

### 3.2. Hemichromes and the Effects of Increased Oxidative Stress on the Erythroid Lineage

Apart from Ab coating, splenic pooling, and failure to upregulate their production through erythropoietin (EPO) increase, red cell life span in VL is further shortened by a cell-disruptive mechanism involving hemichromes [19,20,71]. Hemichromes are denatured hemoglobin chains that result from the oxidation of precipitating free a- and/or b- hemoglobin (Hgb) chains [72,73]. The pathogenesis of the damage they cause has been studied in thalassemias, where unbalanced Hgb subunit synthesis results in clustering, oxidation, and subsequent denaturation of an excess of free hemoglobin chains [72]. Hemichrome accumulation in polychromatophilic erythroblasts, the stage of maximal hemoglobin production, causes ineffective erythropoiesis in thalassemia [72,74].

Experiments in dogs have shown an accumulation of hemichromes in VL [75]. The increased production of reactive oxygen species (ROS) creates a highly oxidative environment that depletes the RBC-reducing potential, leading to the formation of hemichromes [75]. Studies in thalassemia have shown that hemichromes bind to RBC membrane protein band 3, the most abundant RBC integral protein and the major link between the membrane and the cytoskeleton [76,77]. More specifically, they bind to band 3 and release heme iron onto the RBC membrane, forming ROS that alter band 3 conformations, thus exposing its Cys residues to phosphorylation by Syk kinases [77]. Consequently, phosphatases and GSH (reduced glutathione) that naturally comprise the RBC redox armamentarium spontaneously dephosphorylate the protein to its natural state and buffer the released ROS accordingly [77]. Low-grade band 3 phosphorylation, by a small amount of hemichromes, leads to conformational changes in the membrane with band 3 clustering and a release of hemichrome-containing membrane microparticles that rescue the cell from membrane damage and thus erythrophagocytosis [74,77]. Since RBC membrane integrity is the major criterion for the recycling of damaged or aged RBC by the RES, band 3 acts as a major regulator for RBC phagocytosis by communicating the intracellular redox conditions to the membrane [74,76,77]. Under conditions of overwhelming stress, heavy hemichrome accumulation leads to increased band 3 clustering and formation of membrane microdomains that fix autologous circulating Αb and complement C3 fractions, causing increased RBC destruction [74,77].

This model that has been described in thalassemia might also explain RBC destruction in VL, which could be further exacerbated by the concurrent hypergammaglobulinemia. It is also likely that, as in thalassemia, hemichrome accumulation in VL may contribute to limiting the life span and differentiation potential of the maturing erythroid lineage.

### 3.3. Bone Marrow Suppression in Visceral Leishmaniasis

Morphologically, BM aspirates of chronically infected VL patients exhibit marked expansion of the erythroid and myeloid-monocytic lineages, variable suppression of the megakaryocytic lineage, and diffuse plasmacytic infiltration, consistent with the coincident hypergammaglobulinemia [22]. The apparent contradiction between BM hyperplasia and peripheral cytopenias can partially be attributed to the increased destruction of mature blood cells in the periphery. There are also morphologic features indicative of differentiation blockage and dyserythropoietic changes of the erythroid precursors [22]. All these changes constitute ineffective erythropoiesis, which means the increased premature intra BM death of differentiating precursors. Herein, we summarize the updated knowledge regarding the pathophysiologic pathways leading to dysplastic changes in the BM of VL patients. It should be noted, however, that all relevant data have been derived from experimental infections in animals. Therefore, any extrapolation to humans should be attempted with care.

Studies in infected mice have shown effector CD4 T cell accumulation and expansion in a self-stimulatory paracrine loop via intrinsic TNF-α/IFN-γ production and signaling. These cytokines induce loss of stromal cells, with a mechanism similar to that described above for the spleen. Activation and mobilization of hematopoietic stem cells (HSCs) are common in many chronic inflammatory states [23]. Furthermore, during VL, BM stromal macrophages produce GM-CSF and TNF-α that lead to substantial expansion of CFU-GM and to a lesser extent BFU-e/CFU-e in the BM [78]. Despite the expansion of erythroid progenitors, erythropoiesis ceases at the erythroblast stage, followed by differentiation failure and eventual apoptotic death [18]. Apart from heavy hemichrome accumulation, cytokines in the BM microenvironment also significantly disrupt the function of erythroblasts. This process is mediated primarily by INF-γ that silences erythroid differentiation genes and through the upregulation on macrophages of FAS (apoptosis antigen 1), TRAIL (TNF-related apoptosis-inducing ligand), and other apoptotic mediators [79]. The concurrent INF-γ mediated upregulation of their ligands on the surface of erythroid precursors sensitizes them to the pre-apoptotic environment, contributing to their premature intra BM death [79].

Likewise, the mobilization of immature erythroid cells from the BM to the spleen also results in their increased destruction since the splenic environment is extremely hostile for developing erythroblasts in chronic VL.

Regarding the myeloid lineage, the priming and mobilization of differentiating cells by the proinflammatory BM microenvironment, their subsequent relocation in the disturbed splenic parenchyma, and the emergency myelopoiesis are detrimental for parasitic clearance [49,50,78]. Differentiated monocytes are primed by the BM toward a regulatory M2 phenotype [49], which is further consolidated when combined with the effects of HIF1 stabilization in the severely hypoxic VL spleen, as already described above. The combination of the parasite’s proinflammatory effects on the BM and the disorganization of the splenic parenchyma engage the host in a vicious circle of regulatory myeloid cell production that not only is ineffective against the parasite but is used by *Leishmania* for replication.

### 3.4. Bone Marrow Dysplasia in Visceral Leishmaniasis

Eventually, niche disruption and priming of HSCs with GM-CSF and TNF-α produced by bystander BM stromal macrophages allow HSCs to become infected with *Leishmania* amastigotes, which, as has been experimentally shown in vitro, disorganize their differentiation commitment and success (Figure 3) [80].

The HIF1 pathway is manipulated by *Leishmania* in macrophages with a significant impact on the course of the infection. HIF1 is a transcription factor crucial for cellular metabolic adaptation under hypoxic conditions, particularly glucose availability and lipid use. Normally, HIF1 stability is regulated by oxygen tension via its degradation by prolyl hydroxylase 2 (PHD2), a pathway known as the canonical pathway [81]. PKB/Akt, a serine-threonine kinase, directly increases HIF1 gene transcription independently of oxygen concentration. It has been shown that promastigotes directly interfere with the canonical oxygen-sensitive regulation of HIF1 stability by dampening the activity of PHD2, whereas amastigotes can non-canonically (oxygen-independently) stabilize it through activation of the PI3K/Akt axis in different cell types, including macrophages (Figure 3) [38,61].

Most available data on the bioenergetic profile of *Leishmania* spp. derive from their similarities with other members of the kinetoplastid family, especially trypanosomes. Flagellated forms such as the promastigote acquire energy primarily through glycolysis [82,83,84]. Compartmentalization of glycolysis inside a specialized organelle, the glycosome, protects glucose from oxidation inside the mitochondrion and also the cell itself from overactive glycolysis that would consume more ATP than can be produced by it [85,86,87]. In these protozoa, the single mitochondrion plays an essential role in lipid synthesis and the production from glutamate of trypanothione and glutathione (both essential for maintaining the parasite’s redox balance in the mammalian host cells) [84,88,89,90]. Transition to the amastigote stage is marked by a profound downregulation of almost all metabolic pathways to compensate for the nutrient-depleted parasitophorus vacuole (PV) microenvironment [91]. In this metabolic state, the amastigotes protect glucose from degradation and shunt it toward the pentose phosphate pathway for the generation of nucleotides to be used in the division and redox regeneration [91]. In the amastigote, the mitochondrion synthesizes glutamate and glutamine to increase redox potential, using fatty acids that undergo beta-oxidation [91]. The source of these lipids remains unknown and could potentially be the PV itself. After all, the PV membrane is formed by mutual lipid contribution from the host cell and the amastigote [92,93,94]. *Leishmania* infestation also upregulates host cell autophagy and, possibly, the fusion of the autophagosome with the PV leading to increased lipid availability [95,96]. Autophagosomal membranes are mostly formed from the ER by budding, while the PV membrane contains an abundance of ER and lysosomal membrane proteins [97,98]. Subsequently, the sustained activation of the Pi3k/Akt pathway with induction of autophagy and HIF1 stabilization maintain the conditions needed for parasite survival [99].

Research has shown that the PV of several *Leishmania* spp. becomes the focus of continuous Akt activation through a currently uncharacterized pathway [100]. Various PI3K isoforms have been shown to favor *Leishmania* survival, but the exact source of the specific phospholipids on the PV membrane that are modified by PI3K to form Akt anchoring sites remains unidentified [100]. The assumptions that they may be synthesized by the parasite itself or that they may derive from the Golgi apparatus or the plasma membrane have been disproven [100].

HIF1 pathway manipulation in the macrophage favors parasite survival as it increases available glucose, reduces immunologic reaction, and prolongs the life span of the infected cell. Bearing in mind that effective HSC differentiation depends on HIF1 degradation and failure to do so causes MDS changes in the bone marrow [101,102], it is plausible to hypothesize that a HIF1-mediated mechanism might contribute to the myelodysplastic phenomena observed in visceral leishmaniasis and should be studied. Another aspect of the infection that merits further investigation is the possibility of histone acetylation in HSCs that is known to occur in macrophages [103,104].

The effects of BM *Leishmania* parasitism are summarized in Figure 4.

## 4. Conclusions

Clinicians usually encounter VL as an opportunistic infection in patients with cell-mediated immunosuppression or, since the clinical presentation is not specific, in the differential diagnosis of myeloproliferative diseases. Recent advances have enhanced our comprehension of the role the liver, spleen, and bone marrow microenvironments play in the shaping of host-parasite interactions and their defining effect on clinical expression and infection outcome. Thus, in this review, we opted for a presentation focusing mainly on the pathophysiologic processes involved in splenic disorganization and bone marrow dysfunction in the course of VL. Based on the parasite’s biology and the mammalian immune response, we describe how HIF1 and the PI3K/Akt pathway function as major determinants of the observed immune failure. We further explain how microenvironmental changes in the spleen and the bone marrow play a crucial role in shaping a disease-permissive space and have a detrimental effect on bone marrow health and regenerative capacity. Finally, we speculate that manipulation of the PI3K/Akt/HIF1 axis may contribute to the MDS features of VL and merits further investigation.

## Figures and Tables

**Figure 1 microorganisms-09-00759-f001:**
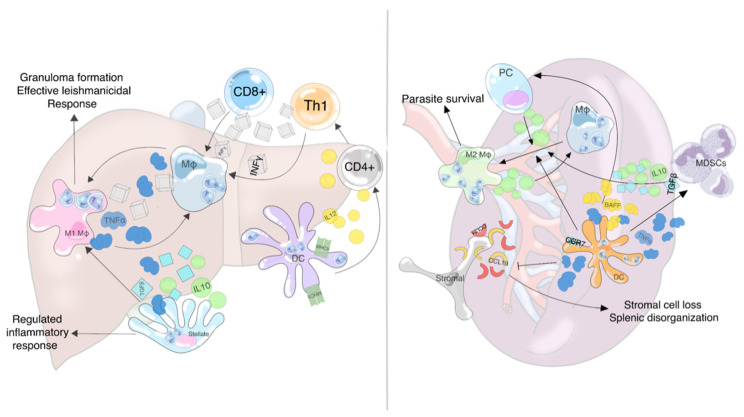
Schematic presentation of the liver (**left**) and spleen (**right**) immune response elicited by leishmanial infection. In the liver, the immune response orchestrated by the liver dendritic cells (DCs) is marked by their releasing interleukin 12 (IL12) that polarizes CD4+ T cells toward the Th1 phenotype. Those along with bystander CD8+ T cells release interferon-gamma (INFγ) that polarize the liver Kupffer cells (liver monocytes-Μφ) toward the M1 leishmanicidal phenotype. The whole immune response dominated by eventual tumor necrosis factor-alpha (TNF-α) production from M1 Μφs is a granulomatous immune response that is adequately balanced from interleukin 10 (IL10) and transforming growth factor-beta (TGFβ) produced from bystander stellate cells. Contrarily, the splenic DCs react to leishmanial infection producing large amounts of TNF-α and B cell-activating factor (BAFF). TNF-α attracts myeloid-derived suppressor cells (MDSCs) that secrete IL10 and TGFβ. Thereafter, the polarization of splenic Mφs toward regulatory M2 Mφ occurs, allowing parasitic survival. BAFF attracts plasma cells (PCs) that also secrete large amounts of IL10. Large amounts of TNF-α also cause downregulation of key stromal cell chemokines, CCL19/CCL21, that regulate homing of T cells and thus splenic architecture. Eventually, further downregulation of CCL19 ligand CCR7 on splenic DCs seals the complete splenic disorganization.

**Figure 2 microorganisms-09-00759-f002:**
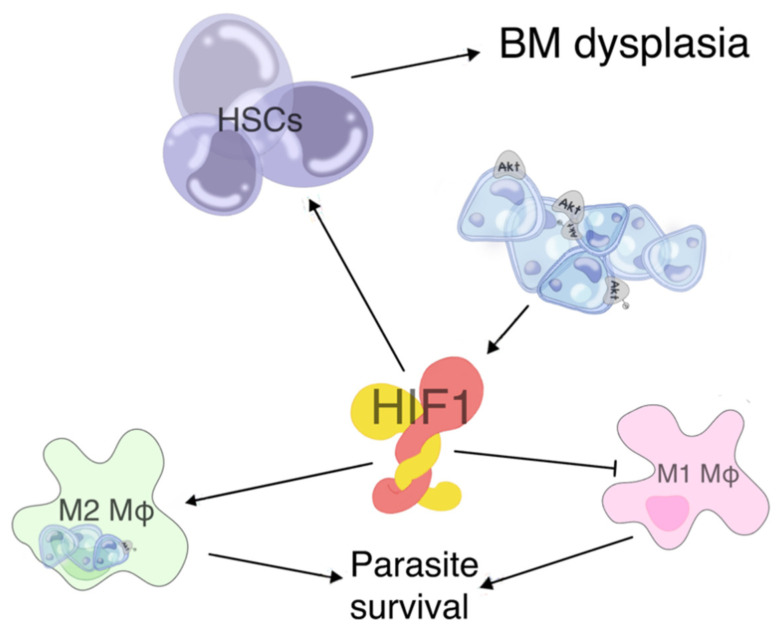
Schematic presentation of the suggested role of hypoxia-inducible factor 1 (HIF1) in visceral leishmaniasis (VL). HIF1 aberrant stabilization either directly following promastigote infestation in the reticuloendothelial host system (RES) cells or secondarily via continuous activation of the PI3K/Akt axis on the amastigote plasma membrane is demonstrated. If the activation occurs in a naive macrophage (Mφ), then the Μφ will acquire a regulatory M2 phenotype instead of an activated M1, thus promoting parasite survival and disease. If HIF1 stabilization occurs in a hematopoietic stem cell (HSC), myelodysplastic phenomena arise that largely contribute to the cytopenias observed in VL.

**Figure 3 microorganisms-09-00759-f003:**
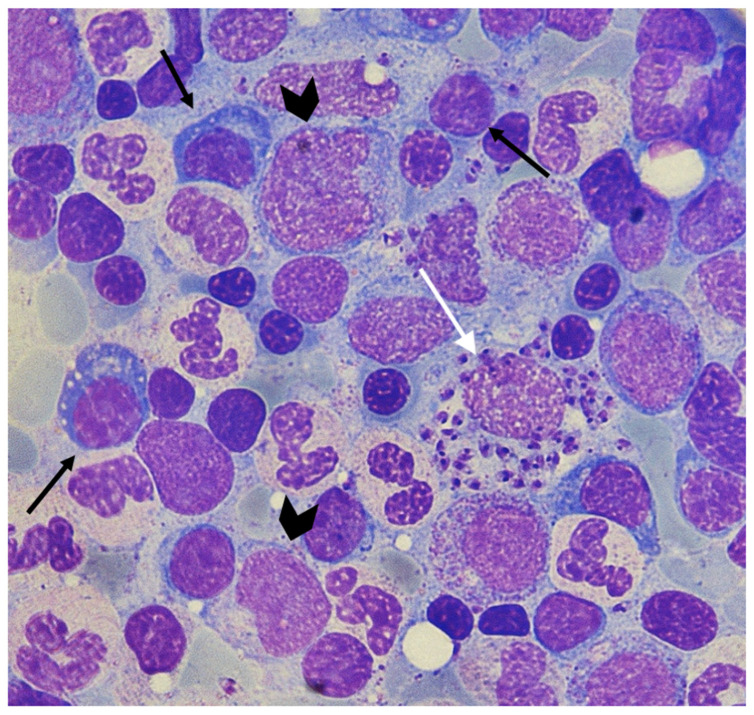
Bone marrow (BM) aspirate from a patient with *L. infantum* infection. *Leishmania* amastigotes are seen within the macrophages (white arrow). Additionally, the patient’s BM exhibits marked dysplasia. Dyserythropoiesis, manifested by asynchronous nuclear-cytoplasm maturation of developing erythroblasts, some with vacuolated cytoplasm (black arrows). Dysgrulopoiesis is also prominent with dysplastic huge agranular metamyelocytes (lower arrowhead) and promyelocytes with abnormal nuclear/cytoplasmic ratio (upper arrowhead) (A.P. personal collection).

**Figure 4 microorganisms-09-00759-f004:**
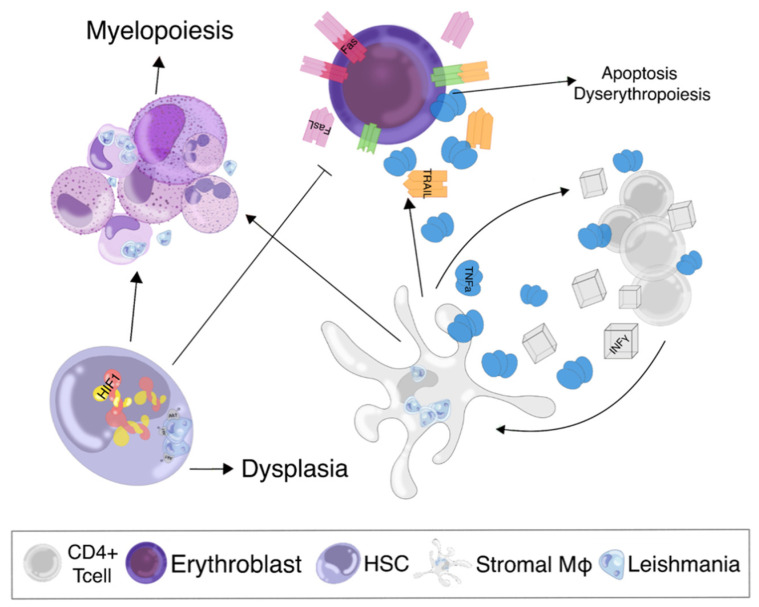
Schematic presentation of the effects of Leishmania infestation on bone marrow (BM). Stromal macrophages (stromal Mφs) infected with the parasite secrete large amounts of tumor necrosis factor-alpha (TNF-α) that stimulate bystander CD4 + T cells to acquire an effector phenotype marked by interferon-gamma (INFγ) release. INFγ further stimulates stromal cells to release TNF-α and forms a self-stimulatory proinflammatory loop. The increased proinflammatory pressure upregulates the proapoptotic molecules released from the BM microenvironment (FasL and TRAIL) along with their receptors on developing erythroblasts (Fas and TRAILr). Furthermore, the proinflammatory microenvironment causes the hematopoietic stem cells (HSCs) to differentiate into the myeloid lineage. Both phenomena cause skewing of hematopoiesis toward myeloids at the expense of erythroid lineage, largely contributing to VL anemia. Furthermore, infection of HSCs leads to Akt docking on the parasitophorus vacuole membrane and thereafter phosphorylation and activation. Activation of Akt further leads to non-canonical, namely oxygen-independent HIF1 stabilization inside the HSCs, which translates to acquired myelodysplasia.

## Data Availability

Not applicable.

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
