# Peer review of "Effects of Visceralising Leishmania on the Spleen, Liver, and Bone Marrow: A Pathophysiological Perspective"

_microorganisms, 2021, doi:10.3390/microorganisms9040759_

Round 1
Reviewer 1 Report
- According to the title and abstract, the pathophysiology during VL will be discussed in the liver and spleen, however it seems like no extensive research has been conducted on the liver. The discussion of the liver only spans from line 92 to 110, whereby all sentences are referred to the same review article from Paul M. Kaye from 2004 (https://doi.org/10.1016/j.cytox.2020.100036). The same reference is also excessively used in the spleen section. The authors should extend their literature research and refer to the original articles.
- From the same lab (Paul M. Kaye) a review was published in 2020 on splenic remodeling during VL, where everything is discussed elaborately. The authors should try to elaborate their review and viewpoints to create sufficient added value on the background of what has already been published.
- In the abstract it is highlighted that the PI3K/AKT pathway will be described, though this is not done thoroughly. For liver and spleen only one sentence (line 200) is written. Only a part in the bone marrow section is reserved for this pathway. Furthermore, a review from 2016 (https://doi.org/10.1016/j.cellimm.2016.09.004), used in this manuscript, already describes this axis in leishmaniasis elaborately.
- (Recent) papers on this pathway that are not discussed e.g.:
- Khadem, F., Jia, P., Mou, Z., Feiz Barazandeh, A., Liu, D., Keynan, Y., & Uzonna, J. E. (2017). Pharmacological inhibition of p110δ subunit of PI3K confers protection against experimental leishmaniasis. Journal of Antimicrobial Chemotherapy, 72(2), 467-477.
- Nandan, C. Camargo de Oliveira, A. Moeenrezakhanlou, M. Lopez, J.M. Silverman, J. Subek, N.E. Reiner (2012). Myeloid cell IL-10 production in response to Leishmania involves inactivation of glycogen synthase kinase-3β downstream of phosphatidylinositol-3 kinase. J. Immunol., 188, 367-378
- Figure 1: no reference + arrows point to different parts than described under the figure.
- References are not correctly added, many consecutive sentences carry the same reference:
- Lines 38 and 39
- 47 and 50
- 197 and 199
- 256 and 259
- 278 and 280
- 321, 323 and 325
- 335, 338 and 340
- Schematic presentations (figure 2 and 3) are very logical and complete and give an added dimension to the review. Minor remark: the size of some words is too little to read properly.
Author Response
Dear reviewer I, please see below our point-by-point responses. We would like to thank you for the constructive criticism and valuable comments toward an improved version of our manuscript. We did our very best to address all comments made aiming to be of benefit to the readership.
Comments and Suggestions for Authors
- According to the title and abstract, the pathophysiology during VL will be discussed in the liver and spleen, however it seems like no extensive research has been conducted on the liver. The discussion of the liver only spans from line 92 to 110, whereby all sentences are referred to the same review article from Paul M. Kaye from 2004 (https://doi.org/10.1016/j.cytox.2020.100036). The same reference is also excessively used in the spleen section. The authors should extend their literature research and refer to the original articles.
Response to comment 1:
We truly appreciate such constructive criticism. Please kindly see the revised version of section 1.1.
including the updated references.
- From the same lab (Paul M. Kaye) a review was published in 2020 on splenic remodeling during VL, where everything is discussed elaborately. The authors should try to elaborate their review and viewpoints to create sufficient added value on the background of what has already been published.
Response to comment 2:
Thank you for the criticism.
Please see revised section 1.2. and revised references.
- In the abstract it is highlighted that the PI3K/AKT pathway will be described, though this is not done thoroughly. For liver and spleen only one sentence (line 200) is written. Only a part in the bone marrow section is reserved for this pathway. Furthermore, a review from 2016 (https://doi.org/10.1016/j.cellimm.2016.09.004), used in this manuscript, already describes this axis in leishmaniasis elaborately.
Response to comment 3:
We truly appreciate your in depth review. Please kindly see revised manuscript especially lines: 252-270
- (Recent) papers on this pathway that are not discussed e.g.:
- Khadem, F., Jia, P., Mou, Z., Feiz Barazandeh, A., Liu, D., Keynan, Y., & Uzonna, J. E. (2017). Pharmacological inhibition of p110δ subunit of PI3K confers protection against experimental leishmaniasis. Journal of Antimicrobial Chemotherapy, 72(2), 467-477.
- Nandan, C. Camargo de Oliveira, A. Moeenrezakhanlou, M. Lopez, J.M. Silverman, J. Subek, N.E. Reiner (2012). Myeloid cell IL-10 production in response to Leishmania involves inactivation of glycogen synthase kinase-3β downstream of phosphatidylinositol-3 kinase. Immunol., 188,367-378
Response to comment 4:
Please kindly see the revised references along with revised manuscript lines: 252-270 and updated references.
- Figure 1: no reference + arrows point to different parts than described under the figure.
Response to comment 5:
Thank you for the comment. Image (now figure 3-line 356) is from A.P.’s personal collection. No reference is used. This is a diagnostic BM aspirate from an infected patient at the hematologic unit where A.P. is occupied.
- References are not correctly added, many consecutive sentences carry the same reference:
- Lines 38 and 39
- 47 and 50
- 197 and 199
- 256 and 259
- 278 and 280
- 321, 323 and 325
- 335, 338 and 340
Response to comment 6:
Thank you for the detailed review. We addressed all references and confirm that they are placed correctly. The manuscript context corresponds to the added reference.
- Schematic presentations (figure 2 and 3) are very logical and complete and give an added dimension to the review. Minor remark: the size of some words is too little to read properly.
Response to comment 7:
Thank you for the comment. Please see revised figures.
Reviewer 2 Report
Overview: Aikaterini Poulaki and colleagues present a review study focused on the pathophysiological alterations associated with the establishment of visceral leishmaniasis (VL), particularly addressing the effect of the disease on liver, spleen and bone marrow. The subject is of scientific interest and importance and supports summarizing information on VL progression, highlighting the events and mechanisms leading to the disruption of normal organ function and onsetting of pathophysiological conditions. A detailed description of VL gradual unfolding in spleen and bone marrow is presented, carefully addressing the disruption of normal tissue physiology and the molecular pathways tackled. Leishmaniasis constitute a group of neglected diseases, that cause a significant disease burden on human populations in some of the poorest parts of the world. VL, constitute a potential life-threating infection, and therefore, the detailed knowledge on the pathophysiology onsetting in the major affected organs is of great importance for the control of the disease progression and cure achievement.
The manuscript is well written, easy to follow and present detailed and updated information on the onsetting of pathophysiological alterations on VL main target organs: liver, spleen and bone marrow. For bone marrow the authors also detail the involvement of HIF1 and the PI3K/Akt pathway in the local immune response. However, there are several points that should be addressed before final acceptance for publication, to improve overall manuscript quality and making it more appealing to the reader.
General comments:
- The authors should revise the Introduction with attention to:
- The information on VL pathophysiology gradual unfolding on host internal organs liver, spleen and bone marrow, are in majority of times derivate from studies in animal models (rodent model) or from canine host, as dogs, like humans, also develop visceral disease. This is clear in the mind of the authors and is detailed latter on the manuscript (ex. Line 134). However, in the first part of the manuscript, namely in the introduction (and also in the description of liver VL associated pathophysiology) is not so clear and can generate some degree of confusion on the reader. Therefore, I suggest a clarification on this, by elucidate the reader on the different sources of information on VL pathophysiology (animal models) and their correlation with what is known for human VL pathophysiology.
- Local tissue/immune response has been pointed as key in VL disease outcome and has particular importance in the context of a detailed analysis of the pathophysiology in each addressed organ, as many of the pathophysiological observations results from the local immune response generated. Although, this is clear in the mind of the authors, the importance of understanding local tissue immune response and local immune interactions should be more emphasized in the introduction, together with the justification of its importance in the context of the disease.
- Section 1. Overview of the leishmanicidal response in the liver should also be revised in order to clarify the complex role of the liver in VL. The liver constitutes a life-supporting organ of the mammal host and is also a target organ in VL. In its present form this section appears to be out of proportion, less developed compared to spleen and bone marrow detailed analysis presented by the authors. Also, it is not clear which of the described events are known to take place in human VL patients and which are derived from experimental animal models. As this manuscript constitutes a revision article, authors may improve their description of the events in the liver related to VL, namely on why is the liver considered to have an effective local immune response to VL, what is precisely the local organ immune response and how it led to granuloma formation and its association with disease control. Likewise, the authors may also briefly elaborate on the impact of the local liver response on VL outcome. Are granulomas helpful in controlling the parasite and avoid VL exacerbation? Does granuloma formation highly impact the liver normal physiology? What happen in the liver if no local granuloma is formed? Addressing these questions, may improve the manuscript quality, providing more complete, clear, and detailed information on liver VL associated pathophysiology.
- As this manuscript constitutes a revision article, the authors could be more forward-looking and incorporate more information, improving overall manuscript quality and reader’s experience. For example, the authors may elaborate on:
- Are VL onsetting events of pathophysiology on the different described organs timely coordinate? For example, does the events that take place in the liver have any influence in splenomegaly and tissue disruption observed in the spleen?
- Adding information on what is known on how the administration of leishmanicidal drugs used in the human treatment of VL (ex.: Sodium stibogluconate, Paromomycin and Liposomal amphotericin B) may impact the pathophysiology observed in spleen, liver, and bone marrow.
- As the authors focused on the HIF1 and the PI3K/Akt pathway in bone marrow local immune response, the presentation of a figure with a schematic representation of the interaction described, summarizing the finding to the reader, may be of interest.
Major issues
Line 83 – Figure 1- I recommend the relocation of this figure to section 2.3 - Bone marrow suppression in visceral leishmaniasis, were it can be properly integrated in the text. In its present location in the manuscript the figure is not well integrated and can be confusing.
Line 89- Please remove this sentence.
Minor issues:
“Leishmaniases” and “Leishmaniasis” – Both words are present in the manuscript. Please revise the manuscript and eliminate “Leishmaniases”, replacing it with “Leishmaniasis”.
Line 14-15 – Please rephrase the first sentence of the abstract, as it is confusing. Suggestion: “Leishmaniasis, constitute a group of neglected parasitic diseases caused by the protozoan genus Leishmania. In humans it can present different clinical manifestations and are usually classified as cutaneous, mucocutaneous and visceral.”
Line 31- Please rephrase the first sentence of the introduction as it is confusing.
Line 60-61- Are not all Leishmania intracellular parasites when in the mammal host? The authors probably intended to refer to unique ability of some Leishmania species to visceralize and disperse inside the mammal host, infecting internal organs. Please rephrase the sentence clarifying the issue.
Line 130- Please correct “the establishment of the infection”.
Author Response
Dear reviewer II, please see below our point-by-point responses. We would like to thank you for the constructive criticism and valuable comments toward an improved version of our manuscript. We did our very best to address all comments made aiming to be of benefit to the readership.
Reviewer II
Overview: Aikaterini Poulaki and colleagues present a review study focused on the pathophysiological alterations associated with the establishment of visceral leishmaniasis (VL), particularly addressing the effect of the disease on liver, spleen and bone marrow. The subject is of scientific interest and importance and supports summarizing information on VL progression, highlighting the events and mechanisms leading to the disruption of normal organ function and on-setting of pathophysiological conditions. A detailed description of VL gradual unfolding in spleen and bone marrow is presented, carefully addressing the disruption of normal tissue physiology and the molecular pathways tackled. Leishmaniasis constitute a group of neglected diseases, that cause a significant disease burden on human populations in some of the poorest parts of the world. VL, constitute a potential life-threating infection, and therefore, the detailed knowledge on the pathophysiology on-setting in the major affected organs is of great importance for the control of the disease progression and cure achievement.
The manuscript is well written, easy to follow and present detailed and updated information on the on-setting of pathophysiological alterations on VL main target organs: liver, spleen and bone marrow. For bone marrow the authors also detail the involvement of HIF1 and the PI3K/Akt pathway in the local immune response. However, there are several points that should be addressed before final acceptance for publication, to improve overall manuscript quality and making it more appealing to the reader.
General comments:
- The authors should revise the Introduction with attention to:
- The information on VL pathophysiology gradual unfolding on host internal organs liver, spleen and bone marrow, are in majority of times derivate from studies in animal models (rodent model) or from canine host, as dogs, like humans, also develop visceral disease. This is clear in the mind of the authors and is detailed latter on the manuscript (ex. Line 134). However, in the first part of the manuscript, namely in the introduction (and also in the description of liver VL associated pathophysiology) is not so clear and can generate some degree of confusion on the reader. Therefore, I suggest a clarification on this, by elucidate the reader on the different sources of information on VL pathophysiology (animal models) and their correlation with what is known for human VL pathophysiology.
Response to General Comment 1: Please kindly see revised manuscript lines: 84-87
- Local tissue/immune response has been pointed as key in VL disease outcome and has particular importance in the context of a detailed analysis of the pathophysiology in each addressed organ, as many of the pathophysiological observations result from the local immune response generated. Although, this is clear in the mind of the authors, the importance of understanding local tissue immune response and
Response to general Comment 2:Please kindly see revised sections 1.1/1.2.
- Section 1. Overview of the leishmanicidal response in the liver should also be revised in order to clarify the complex role of the liver in VL. The liver constitutes a life-supporting organ of the mammal host and is also a target organ in VL. In its present form this section appears to be out of proportion, less developed compared to spleen and bone marrow detailed analysis presented by the authors. Also, it is not clear which of the described events are known to take place in human VL patients and which are derived from experimental animal models. As this manuscript constitutes a revision article, authors may improve their description of the events in the liver related to VL, namely on why is the liver considered to have an effective local immune response to VL, what is precisely the local organ immune response and how it led to granuloma formation and its association with disease control. Likewise, the authors may also briefly elaborate on the impact of the local liver response on VL outcome. Are granulomas helpful in controlling the parasite and avoid VL exacerbation? Does granuloma formation highly impact the liver normal physiology? What happen in the liver if no local granuloma is formed? Addressing these questions, may improve the manuscript quality, providing more complete, clear, and detailed information on liver VL associated pathophysiology.
Response to comment: Please kindly see revised section 1.1.
- As this manuscript constitutes a revision article, the authors could be more forward-looking and incorporate more information, improving overall manuscript quality and reader’s experience. For example, the authors may elaborate on:
- Are VL on setting events of pathophysiology on the different described organs timely coordinate? For example, does the events that take place in the liver have any influence in splenomegaly and tissue disruption observed in the spleen?
Response to 1:
Thank you for your constructive criticism,
Please kindly see the revised manuscript, lines:
- Adding information on what is known on how the administration of leishmanicidal drugs used in the human treatment of VL (ex.: Sodium stibogluconate, Paromomycin and Liposomal amphotericin B) may impact the pathophysiology observed in spleen, liver, and bone marrow.
Response to 2:
Thank you for your in depth review and your interesting questions. We considered your suggestion about adding the existing knowledge on drug and how these alter the host immune response in depth, yet we believe that this would be beyond the scope of the current manuscript, which focuses on the description of the pathophysiology of the VL in spleen/ liver and the BM. Data on the subject of drug/host interaction and the alterations that these cause in the immunopathology of the disease thereafter in the available literature are limited, and are even more scarce with regards to human host.
- As the authors focused on the HIF1 and the PI3K/Akt pathway in bone marrow local immune response, the presentation of a figure with a schematic representation of the interaction described, summarizing the finding to the reader, may be of interest.
Response to 3: Please kindly see figure 2 of the revised manuscript (lines 187-196)
Major issues
Line 83 – Figure 1- I recommend the relocation of this figure to section 2.3 - Bone marrow suppression in visceral leishmaniasis, were it can be properly integrated in the text. In its present location in the manuscript the figure is not well integrated and can be confusing.
Response: Thank you for your constructive suggestions. Figure 1 is now Figure 3, Line 357.
Line 89- Please remove this sentence.
As per the reviewer’s request, we amended our manuscript accordingly. Please note the updated text.
Minor issues:
“Leishmaniases” and “Leishmaniasis” – Both words are present in the manuscript. Please revise the manuscript and eliminate “Leishmaniases”, replacing it with “Leishmaniasis”.
Response: Please kindly note that Leishmaniases is used as the plural of Leishmaniasis.
Line 14-15 – Please rephrase the first sentence of the abstract, as it is confusing. Suggestion: “Leishmaniasis, constitute a group of neglected parasitic diseases caused by the protozoan genus Leishmania. In humans it can present different clinical manifestations and are usually classified as cutaneous, mucocutaneous and visceral.”
Response: Please kindly see the revised lines 14-15.
Line 31- Please rephrase the first sentence of the introduction as it is confusing.
Response: Please kindly note the revised line 31.
Line 60-61- Are not all Leishmania intracellular parasites when in the mammal host? The authors probably intended to refer to unique ability of some Leishmania species to visceralize and disperse inside the mammal host, infecting internal organs. Please rephrase the sentence clarifying the issue.
Response: Please kindly see revised lines 60-61
Line 130- Please correct “the establishment of the infection”.
Response: Please kindly see revised line 130.
Round 2
Reviewer 1 Report
All comments have been adequatly addressed, no further comments on this revised and much improved version.
Author Response
Dear Reviewer,
Thank you for the constructive criticism.
Sincerely,
the authors.
Reviewer 2 Report
Aikaterini Poulaki and colleagues present a review study focused on the pathophysiological alterations associated with the establishment of visceral leishmaniasis (VL), particularly addressing the effect of the disease on liver, spleen and bone marrow.
The authors addressed all previously raised concerns and incorporate new data improving manuscripts overall quality. The authors have detailed the liver immune response to VL, incorporating new data on iNKT cells role in liver granuloma formation and local immune response. Also, a schematic representation HIF1 stabilization effect on spleen host macrophages, as well as more detailed information on PI3K-Akt pathway involvement in spleen pathophysiological onset were added to the manuscript. More information on bone marrow dysplasia in VL were, likewise, incorporated in the revised manuscript. The manuscript has greatly improved with the authors’ revision.
There only minor issues to be addressed, which I detail below:
Line 86- Please remove “2. Results – Discussion”
Line 188 – “RES cells” – please provide a full description for RES for the subtitle of the figure.
Line 192 – “HSC cells” - please provide a full description for HSC for the subtitle of the figure.
Line 248-251 – The number for the bibliographic reference are in superscript. Please correct the format.
Line 358 – “amastigotes are seen within the macrophages.”- please provide an identification (ex.: with arrow) for the amastigotes. Although, they are clear identifiable, but the reader may not an expert in the field.
Author Response
Dear Reviewer,
Thank you for the constructive criticism. All issues have been addressed in the revised manuscript uploaded.
Sincerely,
the authors